# The Effect of Heat Source Path on Thermal Evolution during Electro-Gas Welding of Thick Steel Plates

**DOI:** 10.3390/ma15062215

**Published:** 2022-03-17

**Authors:** Jun Fu, Qing Tao, Xiaoan Yang, Bogdan Nenchev, Ming Li, Biao Tao, Hongbiao Dong

**Affiliations:** 1School of Engineering, University of Leicester, Leicester LE1 7RH, UK; jf320@leicester.ac.uk (J.F.); qt15@le.ac.uk (Q.T.); xy149@le.ac.uk (X.Y.); bn66@le.ac.uk (B.N.); 2Nanjing Iron & Steel United Co., Ltd., Nanjing 210035, China; liming1@njsteel.com.cn (M.L.); taobiao@njsteel.com.cn (B.T.); 3School of Materials and Physics, China University of Mining and Technology, Xuzhou 221116, China

**Keywords:** electro-gas welding, high heat input, heat source movement path, finite element analysis, thermal evolution

## Abstract

In recent years, the shipbuilding industry has experienced a growing demand for tighter control and higher strength requirements in thick steel plate welding. Electro-gas welding (EGW) is a high heat input welding method, widely used to improve the welding efficiency of thick plates. Modelling the EGW process of thick steel plates has been challenging due to difficulties in accurately depicting the heat source path movement. An EGW experiment on 30 mm thickness E36 steel plates was conducted in this study. A semi-ellipsoid heat source model was implemented, and its movement was mathematically expressed using linear, sinusoidal, or oscillate-stop paths. The geometry of welding joints, process variables, and steel composition are taken from industrial scale experiments. The resulting thermal evolutions across all heat source-path approaches were verified against experimental observations. Practical industrial recommendations are provided and discussed in terms of the fusion quality for E36 steel plates with a heat input of 157 kJ/cm. It was found that the oscillate-stop heat path predicts thermal profile more accurately than the sinusoidal function and linear heat path for EGW welding of 30 mm thickness and above. The linear heat path approach is recommended for E36 steel plate thickness up to 20 mm, whereas maximum thickness up to 30 mm is appropriate for sinusoidal path, and maximum thickness up to 35 mm is appropriate for oscillate-stop path in EGW welding, assuming constant heat input.

## 1. Introduction

In the shipbuilding industry and offshore engineering, the size and structures of ships are increasing rapidly [1,2], thus giving rise to a growing demand for higher control in welding thick steel plates [3,4]. Electro-gas welding (EGW) has become an indispensable welding method for shipbuilding enterprises due to its outstanding advantages, such as large heat input and high welding efficiency [5]. As illustrated in Figure 1, EGW is an automatic welding process using a special flux-cored wire with CO_2_ gas protection, and used for the welding of vertical position of steel plates [6]. During the welding process, the torch moves along the chosen weld path from bottom to top. A water-cooled copper slider is placed on the front of the weld, and a ceramic “backing” plate is positioned at the rear. The welding torch is also used to feed the welding metal into the groove. In thick plates welding, the welding pool is restrained by the weld pieces, backing plate and sliding copper shoe, so that single pass welding can be accomplished.

In EGW, heat transfer from the heat source to the plates occurs mainly through radiation and convection [7,8]. In thick steel plate welding, the efficiency of heat transfer in the “y” direction (with depth, Figure 1) of single pass EGW is a key factor affecting the quality of the joint [9,10]. In 2016, Hwang, Kim and Lee [11] introduced double ellipsoidal moving heat sources to model the temperature profile and residual stress distribution in EGW of marine steel. However, the ellipsoidal method oversimplifies the oscillate-stop heat source path of industrial welding process to 1 dimensional linear case, whereby the movement path in the “y” direction is not considered. Therefore, the heat source paths are inconsistent with the real EGW process, hence the predicted fusion lines do not agree with experimental results.

To include an oscillate-stop heat source path within weld modelling, in 2017, Xu, Pan and Wan [12] used a sinusoidal function to simulate metal active gas arc welding (GAW). In 2018, Yuan et al. [13] applied a piecewise function to model the real oscillate movement path of gas metal arc welding. In 2019, da Silva Pereira et al. [14] implemented weave patterns by path parameterization, which improved the accuracy and has been further extended to predict defects in welding [15]. However, the effect of different heat source paths on thermal evolution in modelling EGW of thick steel plates and further optimize heat source paths according to thickness of steel plate has not yet been investigated systematically.

This study investigates the effect of three different heat source paths (linear, sinusoidal and oscillate-stop) through FEA on the resulting thermal evolution in thick steel plate welding. The results for EGW welding of E36 marine steel plates with thicknesses ranging from 20–40 mm and a heat input of 157 kJ/cm are compared based on fusion efficiency. The FEA models are verified by comparing against experimental observations. Finally, practical industrial recommendations are provided for each heat source path’s approach, according to variation in plate thickness.

## 2. Materials and Model Description

### 2.1. Electro-Gas Welding Experiments

This study investigates the effect of heat source path on thermal evolution during EGW of E36 marine steel plates. For the FEA modelling, experimental parameters, such as the geometry of the welding joints, thermo-mechanical and EGW process variables, and steel composition were considered as inputs, listed in Table 1. The temperature variation of material properties such as density and thermal conductivity of the E36 steel plate was also taken into account. JMatPro was implemented to calculate the material properties of the steel [16,17] for the E36 steel composition, shown in Table 2. In addition to the five main elements: carbon, manganese, phosphorus, and sulphur, microalloying elements such as niobium, titanium, and aluminium were added to improve the mechanical properties of steel. The dimensions of the steel plate model are shown in Figure 2. The thickness of the steel plate is 30 mm, while the length and width of the steel plate are 800 mm and 250 mm, respectively.

In the electro-gas welding process of 30 mm thick steel E36, flux-cored wire JIS Z3319 DW-S60G with a diameter of 1.6 mm was used. The chemical composition and the properties of the weld deposit are presented in Table 3 and Table 4.

### 2.2. Three-Dimensional EGW Model for the Numerical Simulation

Heat transfer during welding is a complex process dependent on multiple thermo-physical and geometrical factors. In EGW, heat from the heat source is dissipated to the welding plates mainly by conduction, radiation, and convection. Thus, a balance must be found between the rate of heat generation (speed and energy of EGW) and the rate of dissipation, determined by the case specific geometry and chemistry.

The heat transfer equation is given as follows:(1)$$\rho C_{p}\frac{\partial T}{\partial t}+\nabla \left(-k\nabla T\right)=Q-Q_{r}-Q_{c}$$
where ρ —density of the material; $C_{p}$—specific heat capacity; *T*—absolute temperature; *t*—time; *k*—thermal conductivity; Q—heat source; $Q_{r}$—radiation heat loss; $Q_{c}$—convection heat loss. During the EGW process, the peak temperature of the heat-affected zone can reach more than 2000 °C, thus the temperature gradient between the steel plate and the environment is massive, causing significant radiation. The heat dissipation terms for radiation and convection are:(2)$$-Q_{r}=\varepsilon \sigma \left(T_{amb}^{4}-T^{4}\right)$$
(3)$$-Q_{c}=h_{f}\left(T_{amb}-T\right)$$
where ε—thermal emissivity; σ—Stephan Postman’s constant: 5.67 × 10^−8^ [W/(m^2^·K^4^)]; $T_{amb}$—ambient temperature [K]; *T*—steel plate surface temperature [K]; $q_{r}$—radiant heat transfer flux [W/m^2^];$\;q_{c}$—convective heat transfer flux [W/m^2^]; $h_{f}$—convection heat transfer coefficient between welding parts and the environment.

A three-dimensional EGW model was implemented, as shown in Figure 3a,b. This model uses a free tetrahedral mesh with the following principles designed for improving the computational efficiency:The weld region close to the weld heat source has a larger temperature gradient and, hence, was divided into a finer grid (minimum mesh tetrahedral edge length 7.67 mm).A coarser mesh (maximum mesh tetrahedral edge length 24.24 mm) was applied to the base steel plate areas far away from the heat source, where a smaller change in thermal gradients occurred.

Figure 3b illustrates the thermal profile and fusion line of a steel plate during the EGW process at 520 s at the midsection. The thermal distribution at surface and side of the steel plate is available from the diagram. Experimental fusion lines are used to verify the simulation results. A schematic diagram of the pre-welding bevel and micrograph of real joint after welding is shown in Figure 3c,d, respectively.

The filling of weld metal gradually with the movement of heat source is considered. The study simulates the ‘activate’ and ‘inactive’ of weld metal. Geometry of weld metal is pre-drawn in the work piece gap and properties of weld metal are activate point by point with the heat source movement.

For the welded area of the weld metal, the material properties of the geometry are activated.

For the unwelded area of the weld metal, the material properties of the geometry are inactivated, as shown in Figure 4.

### 2.3. Heat Source Movement Path

#### 2.3.1. Heat Source Model

The earliest and simplest heat source model is the point heat source model proposed by Rosenthal [18] and it has been widely used in welding simulation. Rosenthal’s heat source model applies a quasi-steady state 3D semi-infinite geometry for point source. A velocity term was added to Rosenthal’s model to simulate heat source movement by Lecoanet et al. [19,20,21]. To describe the heat source distribution, a double ellipsoidal heat source model [22,23] was proposed by Rouquette et al., combining two different ellipses, one in the front quadrant and the other in the rear quadrant [24,25]. Laser Welding Processes have been simulated by a double ellipsoidal heat source model in recent years [26,27,28]. The two heat source models, i.e., Gaussian, semi-ellipsoid and double ellipsoidal heat source model, predict similar temperature distribution and distortion [29,30]. A simplified Gaussian heat source model was to improve calculation efficiency by Cai and Norman [31,32].

In this study, the semi-ellipsoid heat source model was selected. The equation of semi-ellipsoid heat source model is shown below:(4)$$q\left(x,\;y,\;z\right)=\frac{6Q}{\pi r^{3}\sqrt{\pi }}\exp\left(-\frac{3\left(x^{2}+y^{2}+z^{2}\right)}{r^{2}}\right)$$
where q(x,y,z) represents the heat flow density distribution of (x,y,z), Q is the effective power of the arc, r is the radius of the semi-ellipsoid heating source. The schematic diagram of semi-ellipsoid heat source is shown in Figure 5.

#### 2.3.2. Heat Source Movement Path

The movement of the heat source is expressed using a linear, sinusoidal and oscillate-stop path. In this study, the movement speed in the z-axis is defined as $v_{1}$ and in the *y*-axis as $v_{2}$. As shown in Figure 6, three welding paths are modelled. Their exact coordinate heat source locations at each time step are expressed in terms of $v_{1}$, $v_{2}$ and then substituted in Equation (4).

1.Linear path heat source

The linear path heat source considers the welding heat source moving in the welding direction with a welding speed of $v_{1}$; the speed of the y-axis direction is zero. The expression of welding heat flux for its movement along the welding direction is shown in Equation (5):(5)$$q\left(x,y,z\right)=\frac{6Q}{\pi r^{3}\sqrt{\pi }}\exp\left(-\frac{3\left(x^{2}+y^{2}+{\left(z-v_{1}t\right)}^{2}\right)}{r^{2}}\right)$$

The simulation of the welding heat source moves from bottom to top along the welding direction with the welding speed during the electro gas welding process. The heat source moves along the centre of the steel plates for different thicknesses, with a welding speed of 6.9 cm/min, as given in Table 1.

2.Sinusoidal path heat source

The heat source, moving both in the welding direction with speed $v_{1}$ and in the thickness direction with speed $v_{2}$, is considered in the sinusoidal path heat source model. The schematic diagram of the path is shown in Figure 6b.

The defined oscillate range is d, the length of the weld beam is L, the welding speed is $v_{1}$, and the oscillate speed of the heat source in the thickness direction is $v_{2}$; the equation of the welding period is:(6)$$T=\frac{2d}{v_{2}}$$

Total welding time:(7)$$t=\frac{L}{v_{1}}$$

The heat source position of the y-axis direction is considered as:(8)$$y=\frac{d}{4}sin\frac{2\pi t}{T}$$

Steel plates with three different thicknesses of 30 mm, 35 mm and 40 mm were used in this study to investigate the applicable thickness for different heat source movement paths. The parameters of the sinusoidal for different thickness steel plates are shown in Table 5.

According to the parameters of the sinusoidal heat source movement path, the position of the heat source changes with time during the welding process, as described in Equation (10). The amplitude of sinusoidal function is 10 mm, 12.5 mm and 15 mm for three thick plates, the period of the sinusoidal cycle is 5.5 s, respectively. The oscillate range is 20 mm, 25 mm and 30 mm. Due to the heat source needing to be close to the surface while leaving a certain distance for technological factors, the distance of 5 mm to the top surface and the bottom surface is appropriate.
(9)$$P_{1}\left(t\right)=A\sin\left(\frac{2\pi t}{T}\right)+k$$

The welding heat flux of the sinusoidal path is described as:(10)$$q\left(x,y,z\right)=\frac{6Q}{\pi r^{3}\sqrt{\pi }}\exp\left(-\frac{3\left(x^{2}+{\left(y-P_{1}\left(t\right)\right)}^{2}+{\left(z-v_{1}t\right)}^{2}\right)}{r^{2}}\right)$$

According to Equation (10), the position of heat source change with time for three thickness plates are shows in Figure 7. The y-axis is the position along the thickness of steel plates and the z-axis is the position along the welding direction from the bottom to the top of steel plates. The movement path of the heat source under three thicknesses is drawn using solid lines with red, blue and green colour and the three thickness of 30 mm, 35 mm and 40 mm steel plates are indicated as dotted lines with red, blue and green colour, respectively. The positions of the heat sources for 1 s, 5.5 s, 6 s, 7 s and 11 s are marked in Figure 7.

3.Oscillate-stop heat source

According to the characteristics of the oscillate-stop path, the heat source not only moves from bottom to top during the welding process but also oscillates in the direction of the depth of the melt pool, which helps the welding bevels on both sides to obtain the same melting depth. According to the welding practice, the oscillate-stop parameters of the heat source for different thickness steel plates are shown in Table 6.

Assume that the oscillate range is *R*, the distance of the oscillate centre to the origin of the coordinates is *C*, the period of the cycle is *T*, cycles is *n*. according to parameters of the oscillate-stop path; the position of heat source relative to the steel plate with different thickness in the welding period is described as Equation (12):(11)$$P_{2}\left(t\right)=\;\{\begin{array}{c} \frac{R}{1.5}t+10-5.5\left(n-1\right)\frac{R}{1.5},\;\;\;\;\;\;\;\;\;\;\;\;\;\;\;\;\;T\left(n-1\right)<t\leq T\left(n-1\right)+1.5\\ C+\frac{R}{2},\;\;\;\;\;\;\;\;\;\;\;\;\;\;\;\;\;\;\;\;\;\;\;T\left(n-1\right)+1.5<t\leq T\left(n-1\right)+2.5\\ -\frac{R}{1.5}t+30-\left[5.5\left(n-1\right)+2.5](-\frac{R}{1.5}\right),\;T\left(n-1\right)+2.5<t\leq T\left(n-1\right)+4\\ C-\frac{R}{2},\;\;\;\;\;\;\;\;\;\;\;\;\;\;\;\;\;\;\;\;\;\;\;T\left(n-1\right)+4<t\leq T\left(n-1\right)+5.5 \end{array}$$

Regarding their thickness, the oscillate centre *C* = 20 mm and period *T* = 5.5 s are constant, so the equation can be simplified to:(12)$$P_{2}\left(t\right)=\;\{\begin{array}{c} \frac{R}{1.5}\left(t-5.5n+5.5\right)+10,\;\;\;\;\;\;\;\;\;\;\;\;5.5n-5.5<t\leq 5.5n-4\\ 20+\frac{R}{2},\;\;\;\;\;\;\;\;\;\;\;\;\;\;\;\;\;\;\;\;\;\;\;\;\;\;\;\;\;5.5n-4<t\leq 5.5n-3\\ -\frac{R}{1.5}\left(t-5.5n+30\right)+30,\;\;\;\;\;\;\;\;\;\;5.5n-3<t\leq 5.5n-1.5\\ 20-\frac{R}{2},\;\;\;\;\;\;\;\;\;\;\;\;\;\;\;\;\;\;\;\;\;\;\;\;\;\;\;\;\;5.5n-1.5<t\leq 5.5n \end{array}$$

Define a function for the cycle of the equation, P(mod(t,5.5)) means the equation cycle one time every 5.5 s. The welding heat flux of oscillate stop path be described as:(13)$$q\left(x,y,z\right)=\frac{6Q}{\pi r^{3}\sqrt{\pi }}\exp\left(-\frac{3\left(x^{2}+(y-P_{2}{\left(mod\left(t,5.5\right)\right)}^{2}+{\left(z-v_{1}t\right)}^{2}\right)}{r^{2}}\right)$$

The position of the heat source changes with time for three thickness plates, as shown in Figure 8.

## 3. Model Verification via Experiment

To verify the accuracy of the model, a cross-section is taken along the thickness (*y*-axis) of the weld joint. As shown in Figure 9, the calculated melting pool and fusion line using different heat source path modes are compared against the experimental welding joints. The model fusion line is taken at the material melting point, 2055 K.

(1)As shown in Figure 9b, the application of the linear heat source path did not cause sufficiently high temperatures to melt the top and bottom of the joint; only the centre part of joint is melted, which differs to the experimental observation.(2)The application of the sinusoidal path (see Figure 9c) leads to a fully melted weld joint in the thickness direction. The fusion line extended to the outside of the welding groove, but the area covered by the fusion line is smaller than that observed in the experiments (see Figure 9a).(3)The application of the oscillate-stop path leads to a fully melted weld joint in the thickness direction. Additionally, the fusion line extended to the outside of the welding groove base metal near the fusion is melted to form a solid joint, and the area covered by the fusion line is similar to that observed in the experiments.

Quantitative analysis is carried out to measure the coordinates of the points of intersection between the fusion lines and top and bottom surface (marked as A, B, C and D in Figure 9). The coordinates for points A, B, C, and D are listed in Table 7. The difference between simulated and experimental coordinates is calculated, defined as “error” as shown in Table 7, where Error = (simulated results-experiment result)/experiment result. The “×” sign refers to the lack of fusion at the given location, i.e., no fusion line in the area.

In the case of the linear heat source path, there are no fusion lines running through the top to the bottom of the welded joint. For the sinusoidal heat source path and the oscillate-stop heat source path, the welding arc not only moves from bottom to top during the welding process but also along with the weld oscillates in the direction of the depth of the melt pool, which helps the welding bevels on both sides to obtain the same melting depth. The “errors” of four selected points with sinusoidal heat source are: 21.4%, 15.5%, 28.6% and 25.0%, respectively, while the error of four selected points with oscillate-stop heat source is 8.8%, 2.0%, 4.8% and 0%, respectively. The “error” of the oscillate-stop heat source is greatly reduced. The heat source with oscillate-stop path holding on at near surface of steel plate for more times than sinusoidal path, which helps to transfer more heat to the surface of the joint during the welding process.

In summary, the path of the heat source can significantly affect the thermal profile of the weld joint, hence FEA models are essential in optimizing and predicting accurately the thermal profile, including the fusion line and shape of the melt zone in different thickness and path conditions.

## 4. Evolution of Thermal Profile in Heat Affected Zone

### 4.1. Simulated Heat Source Paths

Heat source paths calculated using the linear heat source path, sinusoidal path and oscillate-stop path are shown in Figure 10. In the figure, thermal profiles indicating heat sources at 340 s, 349 s, and 357 s are shown as examples. The linear heat source moves from the bottom to the top along the centre of the thickness of the steel plate, the sinusoidal heat source moves along with the weld direction and the thickness of the weld pool at the same time, and the oscillate-stop heat source path stays near the surface for a period of time when oscillating to the internal and external position. The numbers in brackets on the right side of every picture are the coordinates of the heat source centre of every picture. The x axis alone the welding line and the y axis alone is the plate thickness direction. When the welding time is 0 s, the heat source centre is in the starting point of the welding line (z) and the centre of plate thickness (y); the coordinates of the heat source centre are the origin coordinates (0, 0).

### 4.2. Thermal Cycles in Heat Affected Zone

Figure 11a is a schematic diagram indicating the positions of the selected points in the heat affected zone (HAZ). Point 1, point 6 and point 8 are located in the groove which belongs to the molten pool, point 2, point 3, point 4, point 5 and point 7 are located in the base metal, the distance to the heat source of the five points is from 11 mm to 50 mm. Figure 11b–d show the simulated thermal cycle curve of the five selected points during the welding process with the linear path, sinusoidal heat source path and oscillate-stop heat source path, respectively.

As illustrated in Figure 11, the temperature quickly rises when the welding heat source approaches the selected point, reaching a peak value, and then it decreases gradually as the welding heat source moves away from the point. The peak temperature varies according to the distance from the welding central line to the points. As listed in Table 8, the peak temperature of point 1 with a linear heat source path is 1932 K, which does not reach the melting point of 2055 K. The peak temperature of point 1 with sinusoidal heat source path is 2085 K, which exceeds the melting point by 30 K, the point 1 will be melted. The peak temperature of point 1 with the oscillate-stop heat source path is 2156 K, which exceeds the melting point by 106 K, compared with the sinusoidal heat source path; its fusion line advances to a position further away from the groove to form a joint with better quality. The peak temperature values of the selected point are listed in Table 8.

Using the sinusoidal path and the oscillate-stop path, the distance between the heat source and the selected points becomes closer when the heat source moves to the surface. While the oscillate-stop path stays near the internal and external position for a period of time in the cycle of movement. The welding arc can transfer more heat to the surface of steel plates and advance the fusion line in the position. So, the temperature in HAZ with the oscillate-stop path is higher than that with the sinusoidal path. EGW is a single pass method for welding thick steel plates, so the stop-over of the heat source at a near-surface position has great significance on the quality of the weld joint.

## 5. Recommended Heat Source Path Model for Simulating EGW Welding of Thick Steel Plates

Three different heat source paths have been used for simulating EGW welding of thick steel plates. In this section, the study will examine the effect of the heat source model on the shape of weld pool and the fusion line, so that a different applicable heat source model for simulating EGW thick plates can be defined.

The size of the cross section of the work piece gap is shown in Figure 12. The heat source movement path for 20 mm, 25, 30 mm, 35 mm and 40 mm thick steel plate were designed and listed in Table 5 and Table 6 and Figure 7 and Figure 8.

Simulations were carried out on steel plates with different thicknesses and different heat source models, as listed below. Other modelling parameters are listed in Table 3 and Table 4, and the heat input is 157 kJ/cm for all trials.

20 mm, 25 mm thickness steel plates—using the linear path (Figure 12a,b);30 mm, 35 mm thickness steel plates—using sinusoidal path (Figure 12c,d);35 mm and 40 mm thick steel plates—using oscillate-stop path (Figure 12e,f).

For the linear path model: Figure 12a shows a fully fusion welding joint for 20 mm thick steel plates, while Figure 12b shows the lack of fusion in the bottom of the joint for 25 mm thick steel plates. So, the maximum applicable thickness for the linear heat source path is estimated to be 20 mm.

For the sinusoidal heat source path model: Fully fusion of the weld pool was obtained for 30 mm thick steel plates, as shown in Figure 12c, while Figure 12d shows a lack of fusion in the bottom of the joint for 35 mm steel plate. So, the maximum applicable thickness for the sinusoidal function path is estimated to be under 30 mm.

For the oscillate-stop path: Figure 12e shows that a fully fusion of the weld pool is obtained for 35 mm thick steel plates, but the lack of fusion in the bottom of the joint for 40 mm thick steel plates. So, the maximum applicable thickness for the oscillate-stop heat source path is estimated to be around 35 mm.

## 6. Conclusions

Three different types of heat source path models (linear, sinusoidal function and oscillate-stop) were implemented to simulate the EGW process of marine steel with a heat input of 157 kJ/cm.For EGW welding of 30 mm thickness steel plates, the model using the oscillate heat source path predicted a more accurate thermal profile (the shape of melt pool and the fusion line) than those using the sinusoidal and linear heat source paths.The applicable heat source paths for modelling the EGW process of steel plates with different thicknesses were investigated. The linear path model can be used for simulating steel plate thickness up to 20 mm, a maximum thickness of 30 mm is appropriate for the sinusoidal path, and a maximum thickness of 35 mm is appropriate for the oscillate-stop path in EGW welding, with a heat input of 157 kJ/cm.

## Figures and Tables

**Figure 1 materials-15-02215-f001:**
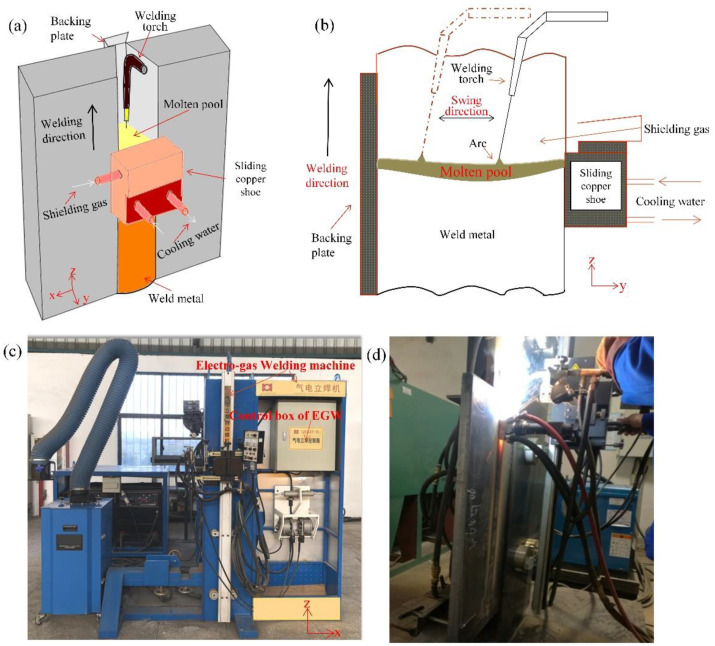
The electro-gas welding (EGW). (**a**) The 3D schematic diagram of EGW equipment, (**b**) The 2D schematic diagram of EGW equipment, (**c**) Photographs of EGW equipment, (**d**) Photographs of experimentation.

**Figure 2 materials-15-02215-f002:**
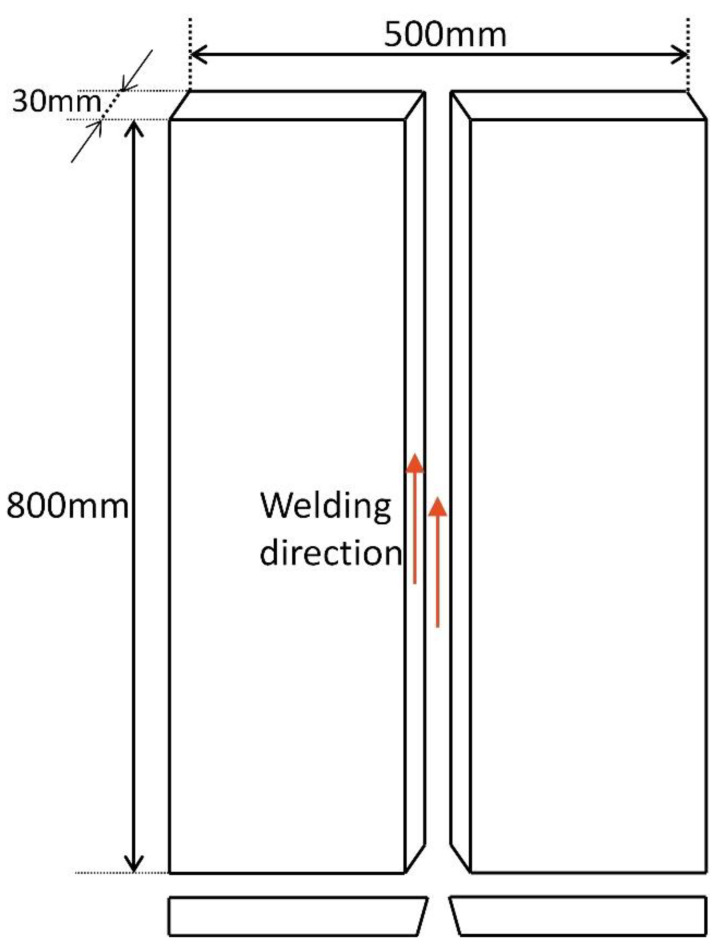
Geometry of welded steel plate.

**Figure 3 materials-15-02215-f003:**
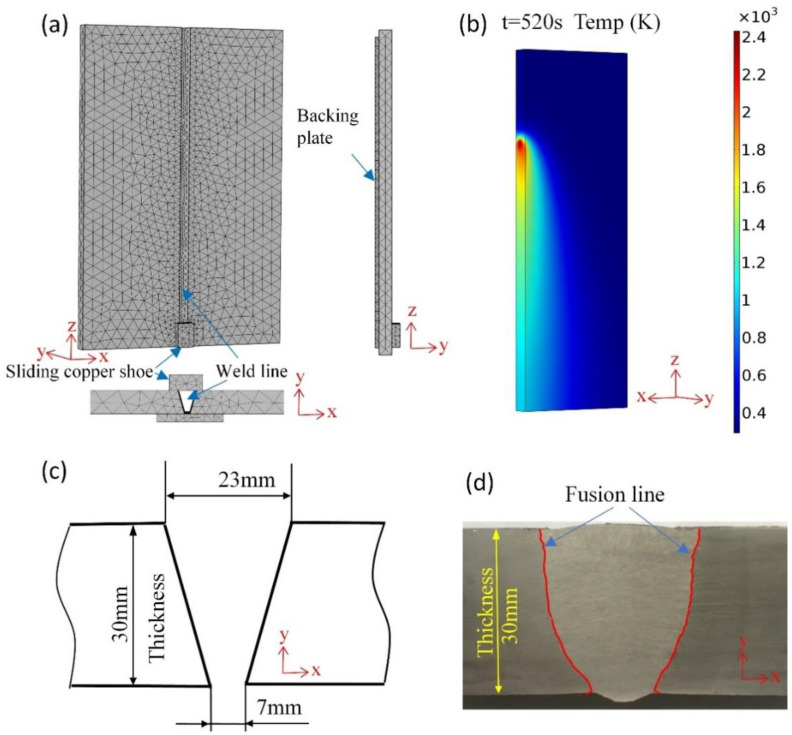
(**a**) Geometry and mesh of EGW model implemented in this study, (**b**) the transient temperature distribution simulated of steel plate at 520 s, (**c**) schematic diagram welding joint of pre-welding bevel, (**d**) welding joint of after-welding showing the fusion line.

**Figure 4 materials-15-02215-f004:**
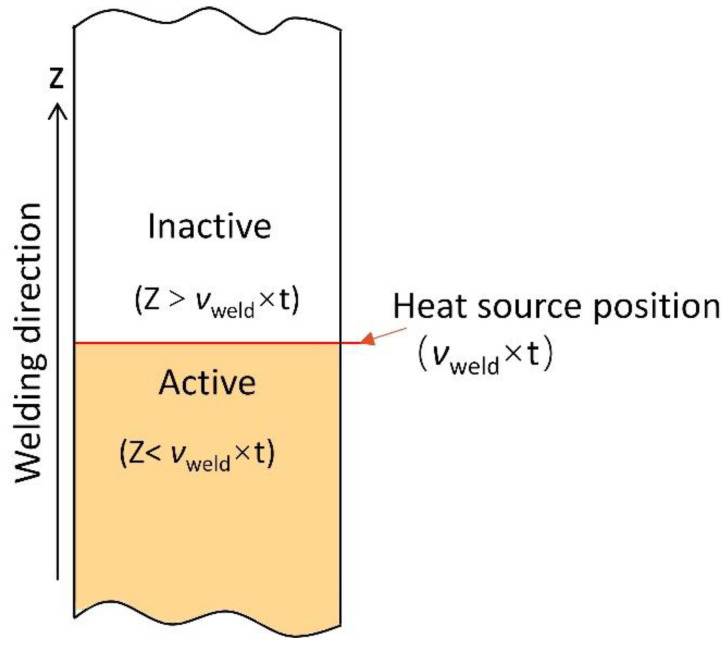
Schematic diagram of the activation process of the weld metal.

**Figure 5 materials-15-02215-f005:**
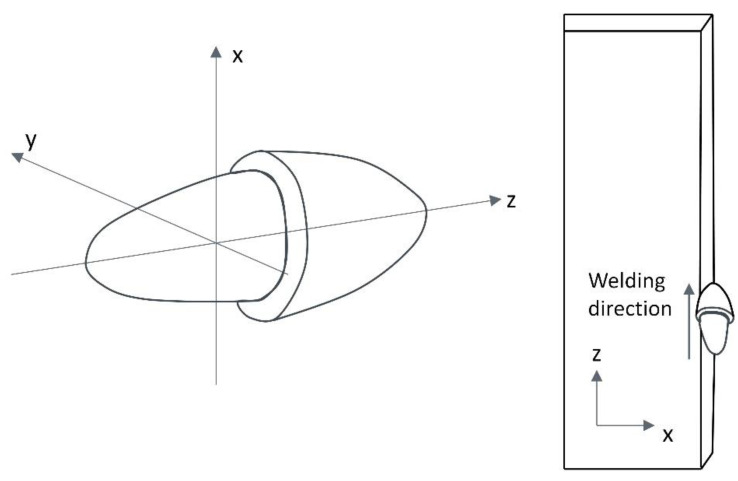
Schematic diagram of heat source (semi-ellipsoid heat source).

**Figure 6 materials-15-02215-f006:**
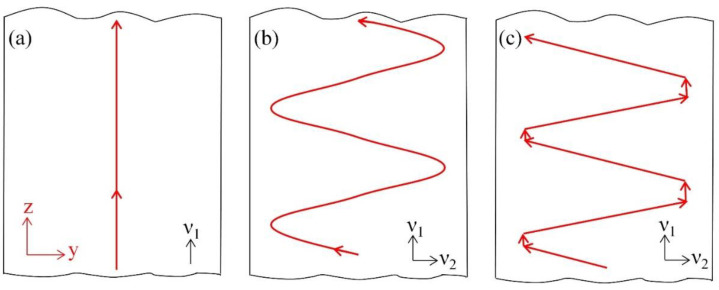
Schematic diagram of heat source path. (**a**) linear heat source path, (**b**) sinusoidal path heat source, (**c**) oscillate-stop heat source path.

**Figure 7 materials-15-02215-f007:**
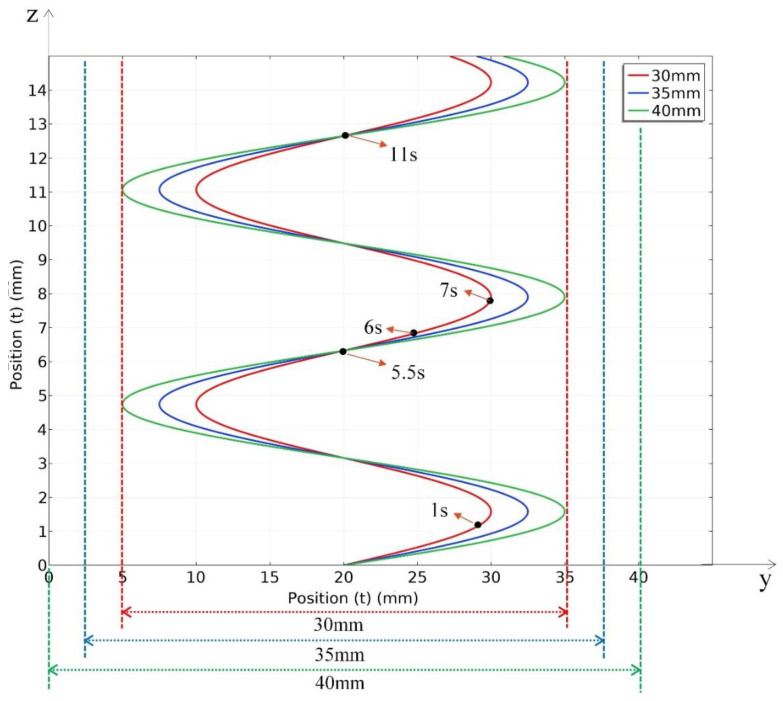
Sinusoidal heat source movement vs. time (t) for 30 mm, 35 mm and 40 mm thick steel plates.

**Figure 8 materials-15-02215-f008:**
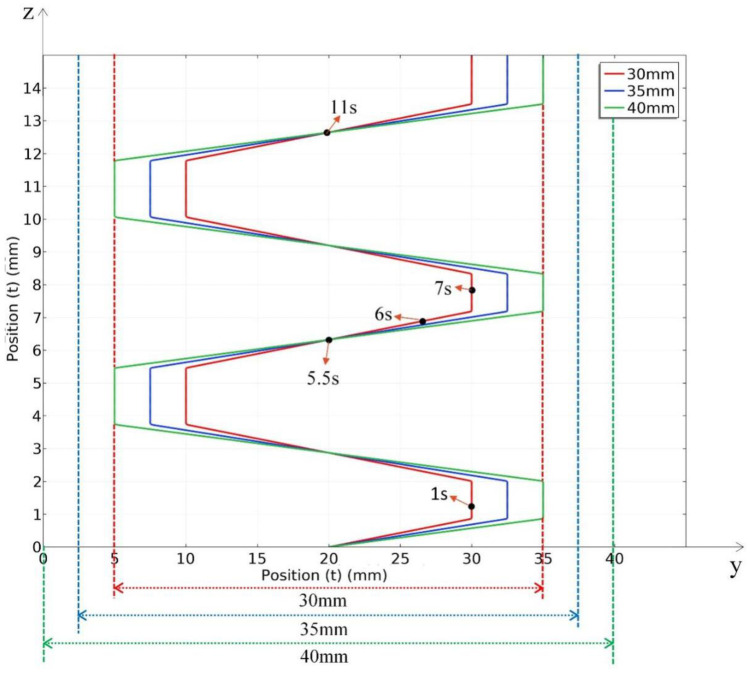
Movement path of oscillate-stop for thickness of 30 mm, 35 mm and 40 mm.

**Figure 9 materials-15-02215-f009:**
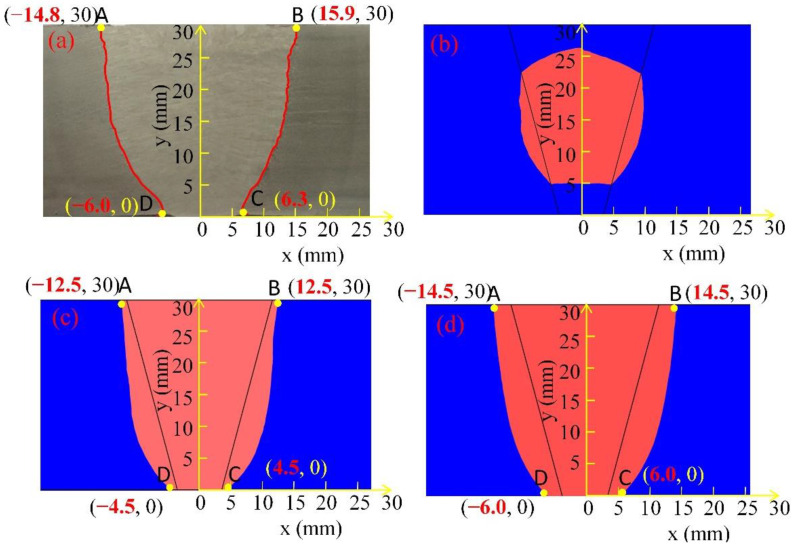
Melt pool and fusion line of experiments and simulated welded joints of 30 mm thickness steel plates, (**a**) experimental weld joint, (**b**) modelling using linear path heat source, (**c**) modelling using sinusoidal path heat source, (**d**) modelling using oscillate-stop path heat source.

**Figure 10 materials-15-02215-f010:**
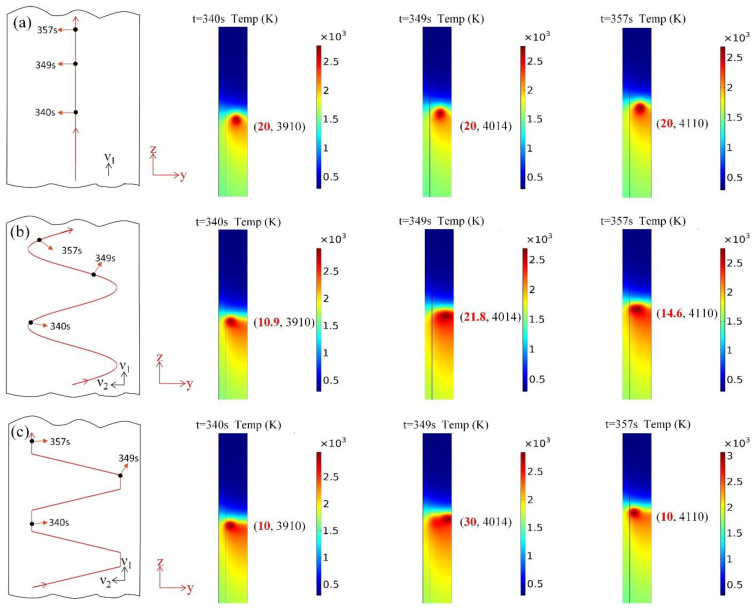
Calculated thermal profiles using three different heat source paths at different welding times of 340 s, 349 s and 357 s. (**a**) linear path (**b**) sinusoidal path, (**c**) oscillate-stop path.

**Figure 11 materials-15-02215-f011:**
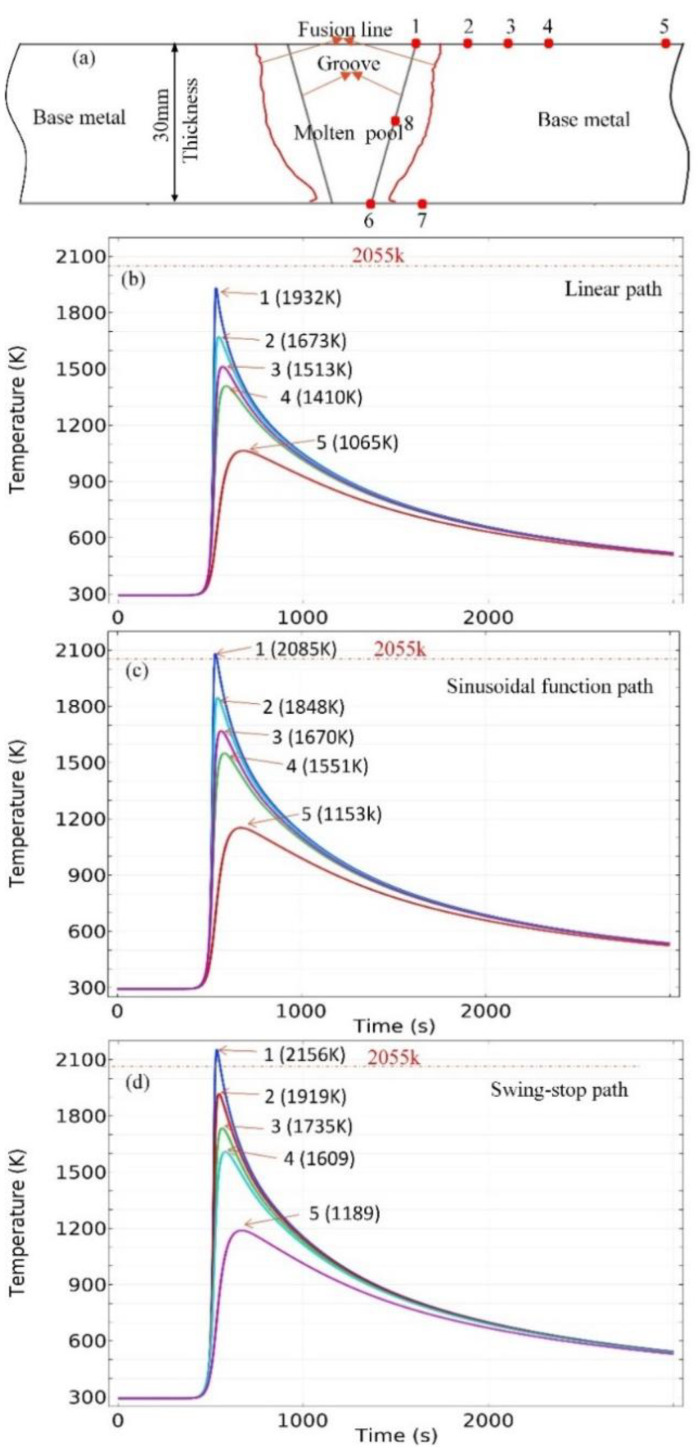
Calculated thermal cycles at selected points in heat affect zones (HAZ), (**a**) position of selected points in HAZ, (**b**) thermal cycle curves using linear path, (**c**) thermal cycle curves using sinusoidal path, (**d**) thermal cycle curves using oscillate-stop path.

**Figure 12 materials-15-02215-f012:**
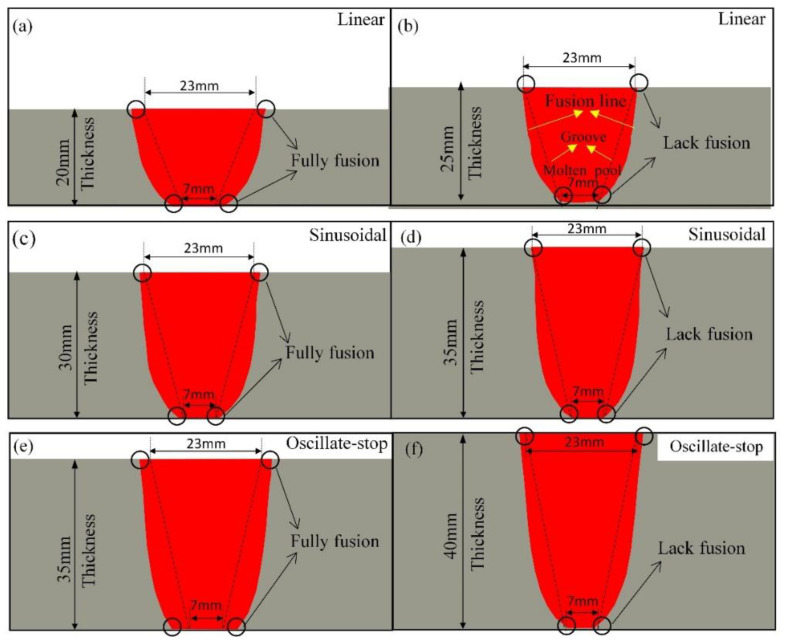
The simulated welding molten pool with three heat sauce paths. Linear heat source path model for 20 mm thickness (**a**), for 25 mm thickness (**b**); Sinusoidal path for 30 mm thickness (**c**), for 35 mm thickness (**d**); Oscillate-stop path for 35 mm thickness (**e**) and for 40 mm thickness (**f**).

**Table 1 materials-15-02215-t001:** Parameters of welding experiments.

Heat Input (kJ/cm)	Welding Speed (cm/min)	Current (A)	Voltage (V)	Wire Diameter (mm)	Wire Feed Rate (m/min)
157	6.9	420	43	1.6	13
Parameters of the torch movement of horizontal oscillations					
Internal stop (s)	External stop (s)	Amplitude (mm)	Frequency (s^−1^)		
1	1.5	10	2/11		

**Table 2 materials-15-02215-t002:** Chemical composition of E36 steel (wt %).

C	Si	Mn	S	P	Nb	Ti	Al
0.08	0.27	1.45	0.002	0.01	0.014	0.014	0.034

**Table 3 materials-15-02215-t003:** Chemical composition of weld deposit.

Grade	C	Mn	Si	S	P	Cr	Ni	Mo	B	Ti
DW-S60G	0.07	1.68	0.33	0.006	0.011	0.02	0.77	0.26	/	0.02

**Table 4 materials-15-02215-t004:** Mechanical properties of weld deposit.

Yield Strength Rp0.2 (MPa)	Tensile Strength Rm (MPa)	Elongation A (%)	Akv at −20 °C (J)
534	662	26	124, 139, 120

**Table 5 materials-15-02215-t005:** Sinusoidal path parameters of welding heat source.

No.	Thickness of Plate (mm)	Amplitude (*A*) (mm)	Period (*T*) (s)	Oscillate Range (mm)	Offset (*k*) (mm)
1	30	10.0	5.5	20	20
2	35	12.5	5.5	25	20
3	40	15.0	5.5	30	20

**Table 6 materials-15-02215-t006:** Oscillate-stop parameters of welding heat source.

No.	Thickness (mm)	Internal Stop (s)	External Stop (s)	Oscillate Range (*R*) (mm)	Oscillate Center (*C*) (mm)	Period (*T*) (s)
1	30	1.0	1.5	20	20	5.5
2	35	1.0	1.5	25	20	5.5
3	40	1.0	1.5	30	20	5.5

**Table 7 materials-15-02215-t007:** Comparison of coordinates of selected points between simulated and experimental welding melt pools.

Coordinate	Experiment (mm)	Linear (mm)	Error	Sinusoidal (mm)	Error	Oscillate-Stop (mm)	Error
A	(−15.9, 30)	×	×	(−12.5, 30)	21.4%	(−14.5, 30)	8.8%
B	(14.8, 30)	×	×	(12.5, 30)	15.5%	(14.5, 30)	2.0%
C	(6.3, 0)	×	×	(4.5, 0)	28.6%	(6.0, 0)	4.8%
D	(−6.0, 0)	×	×	(−4.5, 0)	25.0%	(−6.0, 0)	0

Note: Error = (simulated result—experiment result)/experiment result.

**Table 8 materials-15-02215-t008:** The peak temperature of selected point as shown in Figure 8.

	Point	1 (K)	2 (K)	3 (K)	4 (K)	5 (K)	6 (K)	7 (K)	8 (K)
Path									
Linear path		1932	1673	1513	1410	1065	1810	1640	2420
Sinusoidal path		2085	1848	1670	1551	1153	2021	1811	2295
Oscillate-stop path		2156	1919	1735	1609	1189	2410	1960	2223

## Data Availability

For further data sets, please contact the corresponding author.

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
