# Peer review of "The Effect of Heat Source Path on Thermal Evolution during Electro-Gas Welding of Thick Steel Plates"

_materials, 2022, doi:10.3390/ma15062215_

Round 1

Reviewer 1 Report

The manuscript is on topic in simulation of electro-gas welding (EGW) of thick steel plates. Some experimental results are also presented. The numerical results are well presented. Before publishing, I would advise the authors to consider the following points:

1. It is necessary to clearly describe the experiments performed. It is best to do this in a separate section, and also note it in the abstract.

2. The numerical method for solving the problem is not described at all.

3. Line 56. Mentioned "...in recent years [19-21]". However, [20] is an article from 2009, and [21] from 1998. It is difficult to attribute 1998, and even 2009, to the “recent” category. Similarly, lines 107, 108: "Recently" for [32] (2001), [33] (1998).

4. Table 5. C/Oscillate-stop should be (6.0, 0). Also D/Oscillate-stop should be (-6.0, 0).

5. "Reference" does not contain any links to an article from the journal for which this manuscript is intended. How can this be explained? Are the authors unfamiliar with articles from this journal? Other scientists do not develop this direction or close to it and do not publish in this journal? The topic of the manuscript does not match the topic of the journal?

Reviewer 2 Report

The authors studied the effect of three different heat source path models (linear, sinusoidal and oscillate-stop) on simulating thermal evolution of the welding thick steel plates. While the manuscript is generally well executed, there are several issues that should be addressed before further consideration for publication.

  1. What are the assumptions and boundary conditions used in the models? How are they derived? For example, is heat transferred by radiation ignored?
  2. The error for the sinusoidal source is relatively large, is there any improvement made for a more accurate model? Or this is the optimal? 
  3. Any replicates used in the experiment validations?

Reviewer 3 Report

The authors report on a simulation study of electro-gas welding of thick steel plates. The topic is interesting and falls into the scope of the journal. The authors give a detailed literature review, which confirms that their work is original. However, there are a number of insufficiencies in the work. It starts with an insufficient description of the validation experiments, followed by an inconsistent description of the main issue – the heat source model in the simulation, a limited experimental validation of the simulation and some unexpected results. Therefore, the paper can be maybe published only if the authors can explain or remove all the inconsistencies in the model and revise the paper considerably.

The critical points are:

  1. The welding experiment must be described in sufficient detail. It is given in the introduction, that a flux cored wire is used to fill the gap. Please, give wire diameter, wire feed rate and parameters of the torch movement like amplitude and frequency of horizontal oscillations (in direction of the plate thickness) in table 1. It would be nice to know also the wire composition (e.g. in table 2).
  2. With respect to the model: the grid resolution in Fig. 2a seems to be much coarser (three points over the plate depth near the joint) than given in the text (minimum length 1 mm).
  3. How the material input by the wire melting is introduced in the model?
  4. Please, give a reference of JMatPro or explain the code.
  5. A Gaussian heat source model (equ. (1)) distributes the heat over an area of the melt pool. If the melt pool is horizontal (perpendicular to z direction), the constants A and B in equ. (2) should never be zero even for a non-moving heat source. Identifying the constants A to C with velocities, the heat source model in equ. (2) is difficult to understand: for the simplest case (A=0, v_2=B=0) and a constant welding speed C=v_1 the heat q decreases with z, but why? In contrast, equ. (3) considers the movement in z direction, but the distribution of the heat in x and y direction (as in (1)) is missing. A similar problem arises in eqs. (6), (7) and (11) and also in (15) – the heat distribution in x direction is missing, and it does not correspond to a Gaussian heat source model, where the heat must be distributed over the melt pool surface (in x and y direction). However, a distribution in z direction (perpendicular to the melt pool surface seems me strange.
  6. Equations (8) to (10) and Fig. 4 can be omitted, they give no new information. But the symbols “d” and “T” should be introduced in Table 3, and the three cases are clear. A similar comment holds for the Oscillate-Stop model. It would be sufficient to define a general formular with variables given in Table 4 instead of eqs. (12) to (14).
  7. Results in Fig. 6 are strange. If the authors want to study the impact of the heat source movement, I expect that the heat and wire material amount per welding time should be constant for the same welding speed (to fill the work piece gap). Hence, it is not clear how the gap regions above and below the melted zone in Fig. 6b are filled with material. To understand how the melted zone can be larger in Fig. 6d in comparison with Fig. 6c, the temperature distribution at the welding position should be shown. Furthermore, a comparison with only one experimental case and one joint cross section is insufficient to validate the model and consider a kind of statistical errors.
  8. With respect to Fig. 7, at which position x the profiles are shown? If it is the center of the work piece gap, why the melt pool and the region above the pool (gas or empty) cannot be seen? What are the numbers in brackets on the right side of every picture?
  9. The results in Fig. 8 are not very interesting. As an effect of the moving heat source, a better distribution of the heat over the plate thickness should be expected. Therefore, a comparison of the temperatures in the centre of the plate (middle point between 1 and 6) with those at points 1 and 6 would be useful. Unfortunately, the curves for point 6 and the middle point are not shown. As discussed before, from the results in Table 6 I would expect that the heat input is higher in case of oscillating-stop mode in comparison with the other. But the heat input is the same (157 kJ/cm) in all cases?
  10. The comparison of welding plates of different thickness starting a t page 13 must be explained in more detail. What is the cross section of the work piece gap and what are the torch movement and wire feed parameters? I guess, that the gap cross section is the same in all cases corresponding to the fixed heat input of 157 kJ/cm? Further comparison with experiments would be useful to separate the impact of the torch movement in the simulation and in the real process.
  11. The paper suffers from a large number of grammar errors (e.g. wrong tenses and singular / plural errors). Sentences are not complete and verbs are missing.

Reviewer 4 Report

The researchers have discussed in detail the heat evolution process of electro gas welding. Critical investigations are carried out through simulation of heat evolution during the process of electro gas welding.  The major application areas of the these type of processes are shipbuilding with special reference to offshore engineering.  The researchers have carried out simulation modelling of heat input which is the main source of heat generation. the main concerns of the research is to enhance the welding efficiecy in terms of volume of heat evolved. the researchers adopted a Gaussian heat source to model the generation of heat. Bead geometries are adopted from the industrial sectors to have a pragmatic investigation. Detailed experiments are carried out to verify the theoretical results. Among the different strategies of modelling some particular method of modelling adopting the oscillate-stop heat path is observed to be providing the most predictable results. Based on these study some useful recommendations are made like the value of the heat source for the relevant thisk plate welding process. This is an excellent article but can be published only if the following minor revisions are appropriately adhered to.

  1. The research gap is almost non-existant.
  2. The motivation for the research should evolve spontaneously from the research gaps. As there is hardly any the fixation of the objective of the article is uncertain.
  3. In many situations many of the references are clubbed together for citations. This clubbing activities manifest lack of focus in the extensive literature review.
  4. In various places in the text of the manuscript first person like "our" is used. The events should be described in passive voices without using the first and second person.
  5. Heat source path is shown but the basic geometry of the heat source like double ellipsoidal shapes are not shown.
  6. The word validation is wrongly used. It should be verified as the those theoretical outcomes are not verified in the application indudtries. So it is a classic case of verification but not validation.
  7. Photograpghs of actual experimentation are to added for the documentary evidences of experimentation.

Round 2

Reviewer 2 Report

NA